# MYB Transcription Factors as Regulators of Secondary Metabolism in Plants

**DOI:** 10.3390/biology9030061

**Published:** 2020-03-24

**Authors:** Yunpeng Cao, Kui Li, Yanli Li, Xiaopei Zhao, Lihu Wang

**Affiliations:** 1Key Laboratory of Cultivation and Protection for Non-Wood Forest Trees, Ministry of Education, Central South University of Forestry and Technology, Changsha 410004, China; xfcypeng@outlook.com (Y.C.); yan_li321@163.com (Y.L.); 2Key Lab of Non-wood Forest Products of State Forestry Administration, College of Forestry, Central South University of Forestry and Technology, Changsha 410004, China; 3Science and Technology Promotion Center, Huaihua Forestry Research Institute, Huaihua 418000, China; 13720347@ahau.edu.cn; 4College of Life Sciences, Anhui Agricultural University, Hefei 230036, China; 16710048@ahau.edu.cn; 5College of Landscape and Ecological Engineering, Hebei University of Engineering, Handan 056038, China

**Keywords:** MYB TFs, secondary metabolism, plants transcription

## Abstract

MYB transcription factors (TFs), as one of the largest gene families in plants, play important roles in multiple biological processes, such as plant growth and development, cell morphology and pattern building, physiological activity metabolism, primary and secondary metabolic reactions, and responses to environmental stresses. The function of MYB TFs in crops has been widely studied, but few studies have been done on medicinal plants. In this review, we summarized the MYB TFs that play important roles in secondary metabolism and emphasized the possible mechanisms underlying how MYB TFs are regulated at the protein, posttranscriptional, and transcriptional levels, as well as how they regulate the downstream target gene networks related to secondary metabolism in plants, especially in medicinal plants.

## 1. Introduction

Plant secondary metabolites include many types of compounds, such as flavonoids (phlobaphenes, isoflavonoids, flavanones, flavones, flavonols, proanthocyanidins, anthocyanins), stilbenes, various phenolic acids, and monolignols. The functions of these compounds have been determined and include mechanical strength (cell wall components), signaling molecules, pigments, ultraviolet (UV) light protectants, and phytoalexins. *v-Myb* myeloblastosis viral oncogene homolog (MYB) transcription factors (TFs) represent one of the largest families of a transcription factor in plants. These MYB TFs have important functions in the regulation of the biosynthesis of secondary metabolites in plants.

The MYB TF is named by its conserved MYB domain at the N-terminus and is an evolutionarily conserved present in nearly all eukaryotes [1,2]. *v-Myb* is the first MYB TF identified in avian myeloblastosis virus (AMV) [3]. Subsequently, many MYB TFs have been found in fungi, slime mold, animals, and plants [4,5,6]. Compared with MYB TFs in yeast and animals, the functions and structures of those are conserved in plants [5]. The MYB TF is one of the largest TF families in plants, accounting for about 9% of the total TFs family in *Arabidopsis thaliana* [7]. In *A. thaliana*, researchers have identified more than 1600 TFs, representing about 6% of the whole genome [7,8]. The MYB TF family members typically contain one, two, three, or four imperfect repeats [4,6]. The MYB TFs are classified into four subfamilies that are called 1R-MYB, 2R-MYB, 3R-MYB, and 4R-MYB, based on the structure of the DNA-binding domain. In plants, COLORED1 (C1) is the first gene identified to encode an MYB domain protein, and it has been confirmed to participate in anthocyanin synthesis in the kernels aleurone layer of *Zea mays* [9]. In *A. thaliana*, AtMYB2 shows considerable homology with *C1* from *Z. mays* [10]. The large number of MYB proteins encoded in the plant genome indicates that they may each involve unique functions, which has been confirmed by a large number of published papers [2,11]. The MYB TFs are involved in different biological processes, such as circadian rhythm, defense and stress responses, cell fate and identity, seed and floral development, and regulation of primary and secondary metabolism in plants [2,11,12,13,14]. The secondary metabolites are affected by many factors in medicinal plants [15,16,17]. The MYB TFs also play important roles in the secondary metabolism of medicinal plants [13,18,19,20], which is also an important part of the research and deserves further investigation. Here, we reviewed and discussed the potential functions and emphasized the role and potential regulatory mechanisms of MYB TFs in the secondary metabolism of medicinal plants.

## 2. Diversity and Structure of the MYB TFs

It is now clear that the common characteristic of MYB TFs is the conserved DNA-binding domain (MYB domain), which interacts with DNA [1,2,15]. In general, the MYB domain usually consists of 1-4 imperfect repeats in plants [1,2,15]. Each repeat encodes three α-helices and contains about 50-53 amino acids [1]. Among these three helices, the second and third helices form a helix-turn-helix (HTH) structure [2,21]. The MYB TFs can be inserted into the major groove of the target DNA through the HTH structure and then combine with the target DNA to regulate the expression of the target gene [22]. In general, each MYB repeat domain contains three conserved tryptophan residues, which are separated by 18 to 19 amino acids, forming a secondary structure. According to the number of MYB domains (Figure 1), MYB TF of medicinal plants can be divided into four subfamilies: 1R-MYB, which contains one MYB domain and plays an essential role in regulating plant transcription and maintaining the structure of chromosomes. This subfamily is clustered into several subgroups, including I-box-binding-like, TBP-like, TRF-like, CPC-like, CCA1-like, and other MYB-related proteins [8]. MybSt1 is the first identified MYB-related protein in plants and can be used as a transcriptional activator in potato [23]. As the second-largest subfamily of the MYB TFs, 1R-MYB is widely distributed in plants. There are 64 members in model plants *Oryza sativa,* and 68 members in *A. thaliana* [2]. The *Dendrobium candidum* genome also encodes 42 1R-MYB genes. 1R-MYB TFs involve in flower and fruit development, response to phosphate starvation, circadian clock control, and phenylpropanoid-derived compounds [24].

2R-MYB, which contains two MYB domains, is the most widely existed protein, regulates the transcription of secondary metabolic processes in medicinal plants, and regulates cell differentiation. At the same time, recent studies have shown that 2R-MYB acts as a regulatory protein involved in the metabolic pathways of flavonoids and phenylpropanoids in medicinal plants [20]. The 2R-MYB subfamily, which is probably evolved from an R1-MYB by duplication or from a 3R-MYB ancestor by loss of R1 [5,25], can be grouped into 28 subgroups (Figure 2) on the basis of the conserved amino acid sequence motifs present in their most C-terminal MYB domain and phylogenetic analyses [2,4,26]. There are more than 90 members in the model plants *O. sativa* and 120 members in *A. thaliana* [2]. In medicinal plants, the 2R-MYB subfamily also is the largest group of the MYB family. For example, the *Dendrobium candidum* genome encodes 117 2R-MYB TFs, and the *Jatropha curcas* genome contains 123 2R-MYB TFs [27]. The expansion of 2R-MYB TFs suggests that they may contribute to play a key role in plant-specific processes, which is consistent with the findings of published articles. 2R TFs have been clarified to participate in the determination of cell fate and identity, regulation of development, hormone signal transduction, primary and secondary metabolism, and response to abiotic and biotic stresses. For example, 2R-MYB TFs, such as AtMYB123 (TT2), production of anthocyanin pigment 1–4 (PAP1–4) (i.e., AtMYB114, AtMYB113, AtMYB90, and AtMYB75), VvMYBPA2, VvMYBA2, and ZmC1 regulate proanthocyanidin and anthocyanin pathways [9,28,29,30,31]; AtMYB7 can be downregulated by AtMYB4, which functions as a repressor of flavonol synthesis [32].

The 3R-MYB subfamily, whose members contain three MYB domains, is an evolutionarily conserved group in plants. Genes encoding 3R-MYB proteins have been found amongst plant genomes. There are only five members in model plants *O. sativa* and five members in *A. thaliana* [2]. This phenomenon is also found in medicinal plant genomes. For example, the *D. candidum* genome only encodes four 3R-MYB genes, and the *Jatropha curcas* genome only contains four 3R-MYB TFs. Their role, albeit divergent, is largely related to regulate cell differentiation and participate in cell cycle control [32]. The smallest subfamily is the 4R-MYB group in plants. Each number of 4R-MYB subfamily has four R1/R2 domains. Only one or two 4R-MYB TF has been identified in some plant genomes, for example, only two 4R-MYB members in *A. thaliana* and one member in *O. sativa* [36]. In medicinal plants, the 4R-MYB subfamily is also the smallest group of the MYB family. For example, the *D. candidum* genome only encodes two 4R-MYB TFs, and the *J. curcas* genome only contains one 4R-MYB TF. The function of the members of this 4R-MYB subfamily needs further study.

## 3. Biological Function of MYB TFs

The MYB TFs have a wide range of biological functions and participate in many life processes in plants, such as regulating plant growth and development, involved in cell morphogenesis, regulating primary and secondary metabolic reactions, and responding to abiotic and biotic stresses [22,37,38,39,40,41,42,43]. The MYB TFs also play a very important role in medicinal plants. For example, the *D. candidum* genome encodes 117 2R-MYB genes, and only nine of them are regulated by low temperature. *DoMYB28*, *DoMYB29*, *DoMYB54*, *DoMYB75*, *DoMYB78*, *DoMYB81,* and *DoMYB111* have been up-regulated under salinity stress [20]. Effector-reporter co-expression assays and chromatin immunoprecipitation (ChIP) in *Nicotiana tabacum* has confirmed the binding of *Betula platyphylla BplMYB46* to the E-box, GT-box, and TC-box motifs in the promoters of the superoxide dismutase (SOD), peroxidase (POD), and phenylalanine ammonia-lyase (PAL) genes, which function in secondary wall biosynthesis and abiotic stress tolerance [44]. *Gossypium hirsutum GhMYB109*, a homolog of *AtMYBGL1* in *Arabidopsis thaliana*, is specifically expressed in cotton elongating fibers and fiber cell initials, indicating that this gene may be involved in the development of seed trichome in cotton [45,46]. Huang et al. (2016) confirmed EsMYBF1 as a flavonol-specific R2R3-MYB regulator, which participated in biosynthesis regulation of the flavonol-derived bioactive components in *Epimedium sagittatum* [47]. Heterologous expression of *Dendrobium officinale DoMYB75* in *A. thaliana* can significantly increase seed water-soluble polysaccharide content [20]. *Panax Ginseng PgMYB2*, a nucleus localization protein, may play a key regulatory role in ginsenoside synthesis [48].

## 4. MYB TF Regulation of Secondary Metabolic Pathways

Plants will interact with their living environment during evolution. This result is the secondary metabolite, which can accumulate in the plant, resist the invasion of pathogenic microorganisms, and play an important role in the entire metabolic activity of the plant. Previous studies on the regulation of secondary metabolism by MYB TFs have mainly focused on common crops, fruits, and vegetables, such as soybean, rice, pear, and apple [6,8,49,50]. For example, MYB TFs from *Chrysanthemum morifolium* (CmMYB1), *Leucaena leucocephala* (LlMYB1), *Panicum virgatum* (PvMYB4a), *Eucalyptus gunnii* (EgMYB1), and *Zea mays* (ZmMYB31, ZmMYB42) are able to repress lignin biosynthesis [51,52,53,54,55,56] (Figure 2). In *Malus domestica*, *MdMYB3*, *MdMYBA*, and *MdMYB1* can control the red-pigmented anthocyanins biosynthesis in the peel [57,58,59]. *MdMYB110a* is associated with the red coloration of the fruit cortex during the later stages of fruit maturity, and *MdMYB10* can participate in the biosynthesis of anthocyanins in foliage, flesh, and peel [60,61]. In *Petunia hybrida*, 2R-MYB TFs PHZ (purple haze), DPL (deep purple), and AN2 (anthocyanin2) are responsible for vegetative pigmentation, flower tube bud-blush/venation, and full petal color, respectively [62,63,64]. In lily, the expression of the 2R-MYB TFs LhMYB12-Lat, LhMYB12, and LhMYB6 produces, respectively, tepal splatter-type spot, epal, filament and style pigmentation, tepal, tepal leaf/spot pigmentation [65,66]. In *A. thaliana*, AtMYB111, AtMYB12, and AtMYB11 (Figure 2) are all independently capable of activating the genes encoding flavonol synthase (FLS), flavanone 3-hydroxylase (F3H), chalcone isomerase (CHI), and chalcone synthase (CHS), which together determine the content of flavonol [67,68,69]. In strawberry, FaTTG1, FabHLH3, FaMYB11/FaMYB9 (homologs of TTG1, TT8, and TT2, respectively) form a complex that up-regulates the expression of genes encoding leucoanthocyanidin reductase (LAR), anthocyanidin reductase (ANR), anthocyanidin synthase (ANS), thereby increasing the content of proanthocyanidin [70]. However, there are few reports on the regulation of secondary metabolism by MYB TFs in medicinal plants. The study on the secondary metabolic pathways of MYB TFs regulating medicinal plants by reviewing the literature on MYB TFs is summarized in Table 1. We found that the regulation of MYB on the secondary metabolism of medicinal plants is mostly concentrated in the synthesis of flavonoids and organic acids.

### 4.1. The Role of MYB TFs in the Biosynthesis of Flavonoids Secondary Metabolism

The basic structure of flavonoids is that two benzene rings (A and B) are connected in the middle by a heterocyclic pyran or pyran (with double bond) ring (C) [87,88]. The carbon atoms are identified with “primed numerals” for the B-ring and ordinary numerals for A-ring and C-ring, although a modified number system is used for chalcones. The six major subclasses of flavonoids include the isoflavones (e.g., genistein, daidzein), anthocyanidins (cyanidin, pelargonidin), catechins or flavanols (epicatechin, gallocatechin), flavanones (naringenin, hesperidin), flavonols (quercetin, myricetin), and flavones (e.g., apigenin, luteolin) [87,89]. Each type of flavonoid is further modified, such as rhamnosylation, glucosylation, acylation, methylation, or hydroxylation, leading to the colors and enormous diversity of flavonoids scanned in nature. Flavonoids are a major component of most plant pigments. These compounds are closely related to the attraction of pollinators and the spread of seeds. They also participate in the determination of male fertility, signaling during the formation of nodule nodules, and regulation of auxin transport. Additionally, they help plants resist abiotic and biotic stresses. They are also important as nutritional, medical, and pharmaceutical compounds [90].

The reaction and biosynthesis of flavonoids are shown in Figure 3. In addition to the regulation of key functional genes, the biosynthesis of flavonoids is also regulated by MYB TFs [38,89]. The MYB TFs activate multiple genes in the secondary metabolic synthesis pathway to coordinate their expression, thereby initiating secondary metabolism in medicinal plants. MYB TFs have great significance in the secondary metabolism of flavonoids, the most significant is the discovery of the first related gene *C1* gene in maize [9], and then *P1* is also found to be very significant in maize [91]. Studies have shown that MYB transcription factors can increase the expression of chalcone synthase, chalcone isomer, dihydroflavone alcohol reductase, and other enzyme genes in the metabolic process. The maize *C1* gene mainly controls the color of the aleurone layer and the pigmentation embryo, while *P1* mainly exists in vegetative tissues to regulate the anthocyanin synthesis and accumulation [9]. *C1* has a broader DNA-binding specificity than *P1* in maize. From more similar studies, it can be concluded that MYB TFs play a similar role in the secondary metabolism of medicinal plants. For example, Yuan et al. (2015) found that overexpression of *Scutellaria baicalensis SbMYB8* in tobacco could change the expression level of some flavonoid synthesis-related genes and ultimately regulate the biosynthesis of flavonoids [83]. Li et al. (2017) demonstrated that the *Camellia sinensis CsMYB4a* negatively regulated the synthesis of lignin, phenolic acids, phenylalanine, and flavonoids [92]. The *CsMYB4a* could inhibit the promoter activity of five phenylpropanoid pathway genes (*CsANR*, *CsLAR*, *CsCHS*, *Cs4CL*, and *CsC4H*) and two shikimate pathway genes (*CsAROC* and *CsAROF*) [92]. Wang et al. (2018) confirmed that *CsMYB2* and *CsMYB26* were involved in flavonoid biosynthesis by regulating the expression of *CsF3′H* and *CsLAR* in *Camellia sinensis*, respectively [93]. Huang et al. (2015) cloned and isolated *Epimedium sagittatum’s* two TFs (*EsMYBA1* and *EsMYBF1*) and twelve structural genes. The transcriptional analysis suggested that *EsMYBF1* and *EsMYBA1*, together with *EsTTG1* (a WD40 TF) and *EsGL3* (a bHLH TF), were probably involved in the coordinated regulation of synthesis of the flavonol-derived bioactive components and anthocyanins [81]. Thakur et al. (2020) demonstrated that the MYB family of TFs was an interacting partner of *SlCOS1* (costunolide synthase gene), which catalyzed the final key step of costunolide (sesquiterpene lactone) biosynthesis in *Saussurea lappa* [94]. In *Antirrhinum majus*, *AmMYB305* activates the gene encoding phenylalanine ammonia-lyase (PAL) when it is co-expressed in tobacco protoplasts [95]. Both the *Gentiana triflora* TFs—GtMYBP4 and GtMYBP3—can activate the expression of flavonol biosynthesis genes (such as CHS, ANR, LAR, and ANS) and, when heterologously expressed in *A. thaliana*, increase the content of flavonol [84]. In *A. maju*, VENOSA determines vein-associated anthocyanin patterning, whereas ROSEA1 and ROSEA2 are required for bud-blushed patterns, pale pink, bull’s-eye, and full red [78,96].

### 4.2. The role of MYB TFs in the Biosynthesis of Secondary Metabolism of Organic Acids

MYB TFs are also widely involved in the secondary metabolism of organic acids in medicinal plants. It has been found that the R2R3-MYB TFs of the fourth subfamily are involved in the metabolism of phenylpropanoid as a negative regulator in various medicinal plants. Tamagnone et al. (1998) found that overexpression of *Antirrhinum majus AmMYB308* and *AmMYB330* repressed phenolic acid metabolism in transgenic tobaccos [18]. In *Salvia miltiorrhiza*, *SmMYB39* negatively regulates enzyme activities and transcripts of tyrosine aminotransferase (TAT) and 4-hydroxylase (C4H) [76]. *SmMYB39* acts as a repressor and is participated in the regulation of the rosmarinic acid pathway [76]. Constitutive expression in *Saussurea involucrate*, tomato, and tobacco of the potato genes *PAP1*, *StMTF1*, *StAN1*, as well as that of *AtMYB12*, increases the content of chlorogenic acid [98,99] (Figure 2). Heterologous expression of *A. thaliana AtPAP1* gene in *Salvia miltiorrhiza* can significantly increase the content of salvianolic acid B in transgenic plants. *S. miltiorrhiza SmMYB4* gene can act as a transcriptional repressor to regulate *C4H* and ultimately affect the biosynthesis of rosmarinic acid [77] (Figure 2).

### 4.3. The Role of MYB TFs in the Biosynthesis of Lignins Secondary Metabolism

Lignin is an aromatic heteropolymer mainly derived from the monolignols sinapyl alcohol, coniferyl, and p-coumaryl, which produce syringyl (S), guaiacyl (G), and p-hydroxyphenyl (H) subunits, respectively. Lignin contributes to the rigidity and strength of stems, as well as providing most of the mechanical strength of the plant cell wall. MYB TFs are also widely involved in the secondary metabolism of lignin. For example, MYB83 and MYB46 from *A. thaliana* are direct targets of a cell wall-associated protein AtSND1 (NAM-ATAF-CUC [NAC] domain protein 1), and their induction triggers the expression of MYB58, MYB63, and MYB85, which, in turn, can interact with relevant promoter AC elements of the lignin synthesis genes to up-regulate these genes [100,101,102]. Geng et al., (2020) demonstrated that disruption of MYB20, MYB42, and MYB43 resulted in substantial reductions in lignin synthesis and growth development defects in *A. thaliana* [103]. It has been confirmed that some MYB TFs may be the positive regulators of lignin synthesis. These TFs contain EgMYB2 from *Eucalyptus gunnii* [102], PtoMYB216, PtrMYB20, and PtrMYB3 from Populus spp. [104,105], ZmMYB167 from *Z. mays* [106], PtMYB1 and PtMYB4 from *Pinus taeda* [107,108], and AtMYB75 and AtMYB61 from *A. thaliana* [28,109].

Certain MYB TFs have been implicated as repressors of the monolignol pathway. Karpinska et al. (2004) found that reducing the expression of *PttMYB21a* in poplar would lead to the increase of CCoAOMT transcriptional abundance and lignin content, suggesting that this TF might be a transcriptional inhibitor [110]. We also noted that other MYB TFs from *Chrysanthemum morifolium* (CmMYB1 and CmMYB8) [56,111], *Leucaena leucocephala* (LlMYB1) [55], *Panicum virgatum* (PvMYB4a) [54], *Eucalyptus gunnii* (EgMYB1) [112], *Musa nana* (MusaMYB31) [113], and *Z. mays* (ZmMYB31 and ZmMYB42) [51,52] can inhibit lignin synthesis.

## 5. The Regulation Mechanism of MYB TFs

The regulation network of gene expression is interrelated and mutually restrictive. MYB TF regulates the expression of downstream genes, and its expression is also regulated by upstream genes. For example, WD40 protein TTG1, as an important upstream regulator of MYB TFs, plays an important role in many regulatory pathways in MYB [114,115]. MYB TFs not only regulate functional genes but also regulate other TFs. For example, the *A. thaliana* NAC TFs—NST1/2/3 and VND6/7—are regulated by MYB TFs during secondary wall synthesis [102]. Recently, TFs directly regulated by MYB proteins have also been found, such as *AtMYB66*, which directly regulates *GL2* and *CPC*. In addition, other TFs can directly target *MYB*. For example, AGL15 (AGAMOUS-Like15) can bind to about 29 *MYB* genes and activate its transcriptional expression [116].

MYB TFs can be divided into four different regulatory mechanisms: 1, The regulatory mechanism of protein interaction; 2, The regulatory mechanism of transcriptase; 3, The regulatory mechanism of a redox reaction. 2R-MYB is the most widely distributed MYB TF in plants [6,49]. The second MYB domain of this 2R-MYB protein contains a conserved amino acid residue composed of four cysteines (Cys). Under the condition of oxidation, in order to prevent the DNA domain of MYB TFs from being oxidized, two cysteines (Cys) will be oxidized inside the protein molecules to form the disulfide bond (S-S), so as to ensure the normal physiological activity of medicinal plants [117]. 4, Ubiquitin regulation mechanism. There are many studies on the mechanism of MYB under stress in plants, which can be generally divided into two ways: MYB protein performs its function through ABA-dependent way and some functions through ABA-dependent and non-ABA-dependent ways [118,119,120].

## 6. Conclusions and Perspectives

Functional, structural, and phylogenetic analyses have suggested that there are many homologs of MYB TFs with conserved MYB domains, which have similar activities and functions in divergent plant species. In a given subfamily, the biological function and primary protein structure of most MYB TFs are very conserved among angiosperm species, which means that they are evolved from a common ancestral gene. The MYB TFs play important roles in regulatory networks for flavonoid pathways in plants. As part of transcriptional networks, MYB TFs can be regulated by plant growth regulators, environmental signals, microRNAs, and other TFs. There have been more and more studies on the structure and function of MYB TFs, and it has been found that the functions of MYB TFs tend to be diversified.

To date, little is known about the role of the secondary metabolism in plants, especially in medicinal plants. Further studies should focus on detecting the bHLH-binding and promoter- binding capacity of a greater variety of MYB TFs in medicinal plants. Identifying MYB-binding targets using ChIP-seq would provide additional data on targets on a whole-genome scale. Ultimately, detailed knowledge of MYB TFs will help us to further understand the fine transcriptional regulation of the secondary metabolism in plants, especially in medicinal plants.

## Figures and Tables

**Figure 1 biology-09-00061-f001:**
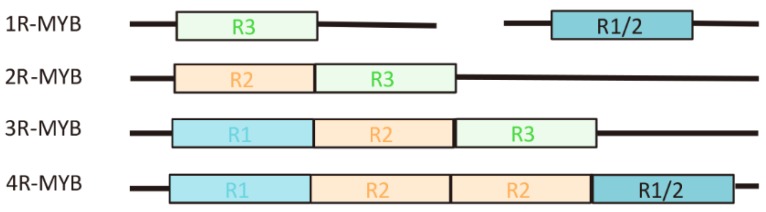
Illustration of classification and structure of MYB TFs in plants. MYB TFs with one to four MYB domain repeats that are identified in plants.

**Figure 2 biology-09-00061-f002:**
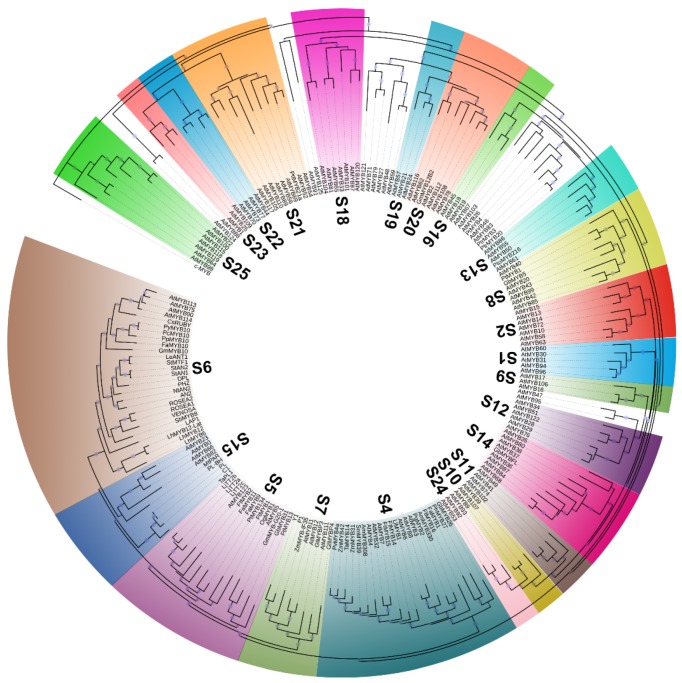
Phylogeny of MYB TFs associated with the secondary metabolic pathway. Multiple sequence alignment of full-length amino acid is carried out using MAFFT 7.0 software (https://mafft.cbrc.jp/alignment/software/) with default parameters. The best substitution model of all amino acid sequences is determined by ModelFinder, and VT+F+R9 is found as the best model fit for these proteins. Maximum likelihood (ML) phylogenetic analysis is performed using IQ-tree software (http://www.iqtree.org/) with an SH-aLRT test for 1000 random addition replicates and a bootstrap test for 10,000 replicates [33,34]. The Figtree software (http://tree.bio.ed.ac.uk/software/figtree/) and iTOL (Interactive Tree of Life) website are used to visualize the phylogenetic trees [35]. The human protein c-MYB is used as an outgroup. The subfamilies (S1–S28) are designated as previously published papers. Accession numbers in DDBJ/EMBL/NCBI databases are as follows: LAP1 (ACN79541); MtPAR (ADU78729); LlMYB1 (ADY38393); LhMYB6 (BAJ05399); LhMYB12 (BAJ05398); LhMYB12-Lat (BAO04194); LjTT2a (BAG12893); LjTT2b (BAG12894); LjTT2c (BAG12895); C1 (AAA33482); PL (AAA19820); PL-BH (AAA33492); P1 (ABM21535); ZmMYB31 (NP_001105949); ZmMYB42 (ADX60106); ZmMYB-IF35 (AAO48737); GmMYB10 (ACM62751); PpMYB10 (ABX79945); CsRUBY (AFB73913); OgMYB1 (ABS58501); PcMYB10 (ABX71487); PyMYB10 (ADN26574); AN2 (AAF66727); DPL (ADW94950); PHZ (ADW94951); PhMYB27 (AHX24372); PtMYB1 (AAQ62541); PtMYB4 (AAQ62540); PtMYB134 (ACR83705); PtrMYB3 (AGT02395); PtrMYB20 (AGT02396); PttMYB21a (CAD98761); PtMYB134 (ACR83705); PtoMYB216 (AFI80906); StAN1 (AGC31676); StAN2 (AAX53089); StMTF1 (ABY40370); TaMYB14 (AEG64799); AmMYB308 (P81393); AmMYB330 (P81395); ROSEA1 (ABB83826); ROSEA2 (ABB83827); VENOSA (ABB83828); Y1 (AAX44239); GmMYB12B2 (AEC13303); GmMYB-G20-1 (BAK24100); FaMYB1 (AAK84064); FaMYB9 (AFL02460); FaMYB10 (ABX79947); FaMYB11 (AFL02461); PvMYB4a (AEM17348); NtAN2 (ACO52470); LeANT1 (AAQ55181); FtMYB1 (AEC32973); FtMYB2 (AEC32975); FtMYB13 (KY290579); FtMYB14 (KY290580); FtMYB15 (KY290581); FtMYB16 (KY290582); SbMYB8 (KF008657); TaPL1 (AK358937); SmMYB39 (KC213793); GbMYB5 (KY703716); GbMYB26 (KY703737); GbMYB31 (KY703742); GbMYBFL (KY678611); PgMYB2 (API61854.1); GtMYBP3 (AB733016); GtMYBP4 (AB289446); c-MYB (X52125).

**Figure 3 biology-09-00061-f003:**
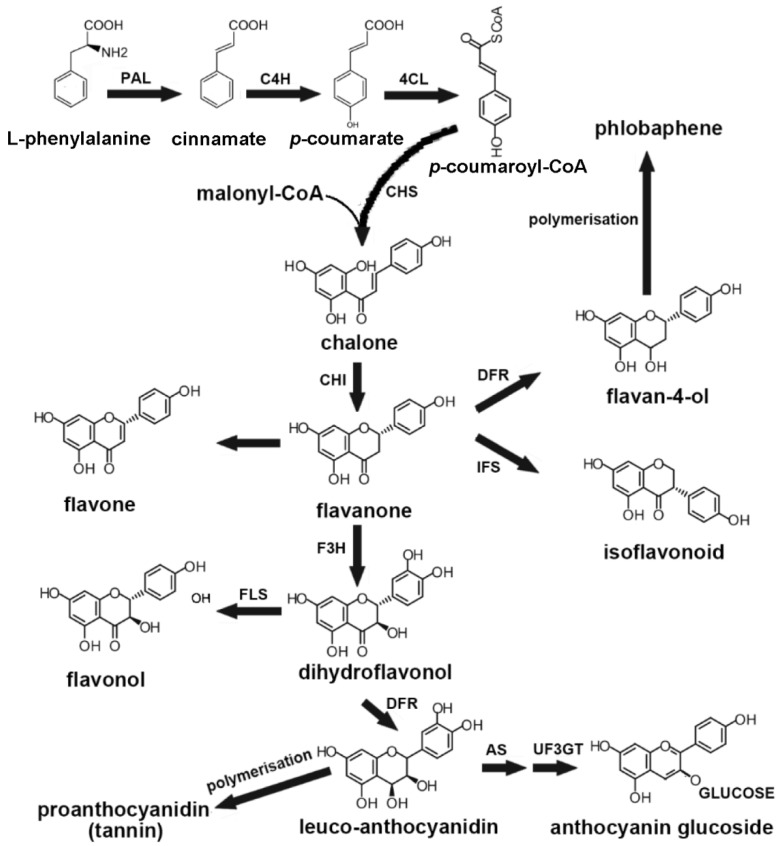
Simplified pathways involved in the biosynthesis of flavonoids in plants. The biosynthetic pathway is adapted from previous studies [88,89,97]. PAL, phenylalanine ammonia-lyase; C4H, cinnamate 4-hydroxylase; 4CL, 4-coumaric acid:coenzyme A ligase; CHI, chalcone flavanone isomerase; F3H, flavanone 3-hydroxylase; CHS, chalcone synthase, DFR, dihydroflavonol 4-reductase; AS, anthocyanin synthase; IFS, isoflavonoid synthase; FLS, flavonol synthase; UF3GT, UDP-glucose: flavonoid 3-0-glucosyl-transferase.

**Table 1 biology-09-00061-t001:** MYB TFs involved in secondary metabolism in plants.

Plant Name	Transcription Factor	Biological Functions	References
*Fagopyrum tataricum*	FtMYB1, FtMYB2,	Overexpression enhances the synthesis and accumulation of anthocyanins	[19]
	FtMYB13, FtMYB14, FtMYB15, FtMYB16	Inhibition of biosynthesis of rutin	[71]
	FtMYB123L	Regulation of flavonoid biosynthesis	[72]
	FtMYB11	Repress phenylpropanoid biosynthesis	[42]
*Lilium brownii var*	LhsorMYB12	Participate in the synthesis of anthocyanidins in leaves	[66]
*Arabidopsis thaliana*	AtMYBL2	Inhibition of anthocyanin synthesis	[73]
	AtMYB21, AtMYB24	Promote the accumulation of anthocyanins	[74]
	AtMYB3, AtMYB4, AtMYB32	Inhibit the accumulation of anthocyanins	[29]
	AtMYB34	Regulation of the synthesis of glucosinolates	[75]
*Triticum aestivum*	TaPL1,	Enhance the synthesis and accumulation of anthocyanins	[43]
*Salvia miltiorrhiza*	SmMYB39	Inhibit the accumulation of phenolic acids	[76]
	SmMYB4	Affect the biosynthesis of rosmarinic acid	[77]
*Antirrhinum majus*	AmMYB308, AmMYB330	Reduce phenolic acid	[18]
	Rosea1, Rosea2 and Venosa	Regulation of anthocyanin production	[78]
*Ginkgo biloba*	GbMYB5, GbMYB26, GbMYB31	Can participate in the biosynthesis of flavonoids under adverse conditions	[79]
	GbMYBFL	Enhance the accumulation of flavonoids and anthocyanin	[80]
*Dendranthema morifolium*	DmMYB1	Negative regulation of the synthesis of flavonoids	[56]
*Panax Ginseng*	PgMYB2	Play a key regulatory role in ginsenoside synthesis	[48]
*Epimedium sagittatum*	EsMYBF1	Increased flavonol content and the decreased anthocyanin content in flowers	[47]
	EsMYBA1	Regulate anthocyanin biosynthesis	[81]
	EsAN2	Regulate anthocyanin biosynthesis	[82]
*Scutellaria baicalensis*	SbMYB8	Regulate flavonoid biosynthesis	[83]
*Gentiana triflora*	GtMYBP3, GtMYBP4	Activate the expression of flavonol synthesis genes	[84]
*Perilla frutescens*	MYB-P1	Determined factor of the anthocyanin forma	[85]
*Medicago truncatula*	MtPAR	Regulate proanthocyanidin (PA) biosynthesis	[86]

Note. The subfamilies and accession numbers of table MYBTFs are given in Figure 2.

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
