# Peer review of "MYB Transcription Factors as Regulators of Secondary Metabolism in Plants"

_biology, 2020, doi:10.3390/biology9030061_

Round 1
Reviewer 1 Report
The sentences starting on line 35 and on line 42 appear to be redundant (or read that way). Can you make it more clear what the point of each sentence is.
Line 59: TF's interact with DNA, not combine
Line 60: combine with the previous sentence.
Line 85: why the parenthesis?
Line 86: add 'the' between in and model
Overall, the review seems well written and put together. I only identified a couple small grammatical errors.
Author Response
The sentences starting on line 35 and on line 42 appear to be redundant (or read that way). Can you make it more clear what the point of each sentence is.
Reply: We sincerely thankful for the careful review from the reviewer. We rewritten these two sentences and removed the redundant parts in the revised manuscript. (lines: 41-42)
Line 59: TF's interact with DNA, not combine
Reply: Following the reviewer’s suggestion, we have revised “combines” to “interacts” in the revised manuscript.
Line 60: combine with the previous sentence.
Reply: Many thanks for this comment. We have revised these sentences.
Line 85: why the parenthesis?
Reply: Many thanks for this comment. We have removed this parenthesis.
Line 86: add 'the' between in and model
Reply: This is a very good suggestion. According to the reviewer’s suggestion, we have added 'the' between in and model.
Reviewer 2 Report
The review article presents a summary of literature on the properties of MYB transcription factors as regulators of secondary metabolism in plants, including medicinal plants. The authors are focused on flavonoids and organic acids secondary metabolism only. This review would be, in my opinion, a much more interesting if it describes MYB TF role in secondary metabolism in all type of plants, especially that a lot of the information contained here concerns to the Arabidopsis and Maize – non-medicinal plants. The other secondary metabolism pathways than flavonoids and organic acids biosynthesis should be present too in this review. In my opinion this manuscript requires revision.
Some important information are omitted in this review, e.g.
AmMYB305 from Antirrhinum majus activated the gene encoding phenylalanine ammonia lyase (PAL) - the first enzyme of phenylpropanoid metabolism (Plant Cell. 1993; 5:1529–1539).
MYB family of TFs is an interacting partner of SlCOS1 gene (costunolide synthase gene) which catalyses the final key step of costunolide (sesquiterpene lactone) biosynthesis in Saussurea lappa (Int J Biol Macromol. 2020;150:52-67).
Two R2R3-MYB TFs (CsMYB2 and CsMYB26) involved in flavonoid biosynthesis in Camellia sinensis tea plant were described in BMC Plant Biol. 2018;18:288
The scheme of flavonoid biosynthesis pathway should be completed. No names of the compounds, e.g. L-phenyloalanine, t-cinnamic acid, ect.
Minor comments:
1) Table1. Arabidopsis thaliana is not medicinal plant. Please changed medicinal plants by plants in the title and heading of table.
2) The names of maize genes: CI gene and Pl are used interchangeably with C1 and P1, respectively (line 2030 208). Please correct.
3) Line 212 please add that CsMYB4a is from Camellia sinensis.
4) Line 251 Please add that NAC TFs NST1/2/3 and VND6/7 are from Arabidopsis.
5) English should be improved by native speaker.
6) The reference list is not prepared correctly. This makes it difficult to use it. References should be described according to the pattern given by the editors ( Author 1, A.B.; Author 2, C.D. Title of the article. Abbreviated Journal Name Year, Volume, page range). Please enter the Abbreviated Journal Names correctly.
E.g. Martin, C.; Paz-Ares, J.J.T.i.G. MYB transcription factors in plants. 1997, 13, 67-73. Please change to Martin, C.; Paz-Ares, J.J. MYB transcription factors in plants. Trends Genet.1997, 13, 67-73.
Author Response
The review article presents a summary of literature on the properties of MYB transcription factors as regulators of secondary metabolism in plants, including medicinal plants. The authors are focused on flavonoids and organic acids secondary metabolism only. This review would be, in my opinion, a much more interesting if it describes MYB TF role in secondary metabolism in all type of plants, especially that a lot of the information contained here concerns to the Arabidopsis and Maize – non-medicinal plants. The other secondary metabolism pathways than flavonoids and organic acids biosynthesis should be present too in this review. In my opinion this manuscript requires revision.
Reply: The reviewer raised a professional and valuable suggestion. According to the reviewer’s suggestion, we have presented a summary of literature on the properties of MYB TF role in secondary metabolism in all type of plants, especially in medicinal plants. Additionally, we also have added other secondary metabolism pathways than flavonoids and organic acids biosynthesis. Kindly see the revised version of the manuscript. (Lines:223-227 and Lines:254-275)
Some important information are omitted in this review, e.g.
AmMYB305 from Antirrhinum majus activated the gene encoding phenylalanine ammonia lyase (PAL) - the first enzyme of phenylpropanoid metabolism (Plant Cell. 1993; 5:1529–1539).
Reply: Many thanks for this comment. We have added this part in the revised manuscript. (Lines: 46-47)
MYB family of TFs is an interacting partner of SlCOS1 gene (costunolide synthase gene) which catalyses the final key step of costunolide (sesquiterpene lactone) biosynthesis in Saussurea lappa (Int J Biol Macromol. 2020;150:52-67).
Reply: According to the reviewer’s suggestion, we have added this part in the revised manuscript. (Lines: 223-227)
Two R2R3-MYB TFs (CsMYB2 and CsMYB26) involved in flavonoid biosynthesis in Camellia sinensis tea plant were described in BMC Plant Biol. 2018;18:288
Reply: Following to the reviewer’s suggestion, we have added this part in the revised manuscript. (Lines: 217-219)
The scheme of flavonoid biosynthesis pathway should be completed. No names of the compounds, e.g. L-phenyloalanine, t-cinnamic acid, ect.
Reply: This is a very good suggestion. We have completed the scheme of flavonoid biosynthesis pathway (Figure 3).
Minor comments:
1) Table1. Arabidopsis thaliana is not medicinal plant. Please changed medicinal plants by plants in the title and heading of table.
Reply: Many thanks for this comment. We have changed medicinal plants by plants in the title and heading of table. Kindly see the revised version of the manuscript.
2) The names of maize genes: CI gene and Pl are used interchangeably with C1 and P1, respectively (line 2030 208). Please correct.
Reply: We sincerely thankful for the careful review from the reviewer. We have corrected these names (Lines: 208-210).
3) Line 212 please add that CsMYB4a is from Camellia sinensis.
Reply: Following to the reviewer’s suggestion, we have added that CsMYB4a is from Camellia sinensis. Kindly see the revised version of the manuscript.
4) Line 251 Please add that NAC TFs NST1/2/3 and VND6/7 are from Arabidopsis.
Reply: Many thanks for this comment. We have added that NAC TFs NST1/2/3 and VND6/7 are from Arabidopsis.
5) English should be improved by native speaker.
Reply: According to your suggestion, we paid special attention to English grammar mistakes. Our paper has been polished by a native English speaker. Kindly see the revised version of the manuscript.
6) The reference list is not prepared correctly. This makes it difficult to use it. References should be described according to the pattern given by the editors ( Author 1, A.B.; Author 2, C.D. Title of the article. Abbreviated Journal Name Year, Volume, page range). Please enter the Abbreviated Journal Names correctly.
E.g. Martin, C.; Paz-Ares, J.J.T.i.G. MYB transcription factors in plants. 1997, 13, 67-73. Please change to Martin, C.; Paz-Ares, J.J. MYB transcription factors in plants. Trends Genet.1997, 13, 67-73.
Reply: Many thanks for this comment. We have corrected all references in the revised manuscript. Kindly see the revised version of the manuscript.
Reviewer 3 Report
General Comments- This manuscript presents a compilation of articles that present information about the MYB Transcription Factors as Regulators of 2 Secondary Metabolism in Medicinal Plants. The topic is timely, although I have a concern over the title about medicinal plants. Because most of the review articles is on plants that are not considered medicinal as Zea mays, Triticum aestivum between others. For that reason, I suggest renaming the article and some sections. for the importance that MYB Transcription Factors have in the secondary metabolism.

Author Response
This manuscript presents a compilation of articles that present information about the MYB Transcription Factors as Regulators of 2 Secondary Metabolism in Medicinal Plants. The topic is timely, although I have a concern over the title about medicinal plants. Because most of the review articles is on plants that are not considered medicinal as Zea mays, Triticum aestivum between others. For that reason, I suggest renaming the article and some sections. for the importance that MYB Transcription Factors have in the secondary metabolism.
Reply: The reviewer raised a professional and valuable suggestion. According to the reviewer’s suggestion, we have renamed the article and some sections. Additionally, our paper has been polished by a native English speaker. Kindly see the revised version of the manuscript. Kindly see the revised version of the manuscript.
Round 2
Reviewer 2 Report
The revised manuscript “ MYB Transcription Factors as Regulators of Secondary Metabolism in Plants” has been improved by the authors, the comments raised by the reviewer have been addressed.
The authors have added a new section “The role of MYB TFs in the biosynthesis of lignins secondary metabolism” to the manuscript, which increases the value of the paper.
In my opinion the revised manuscript is acceptable for publication.
Minor comment - Please changed medicinal plants by plants in the Table 1.